# Development of a Scale to Measure Healthy Behaviors in Spanish-Speaking University Students

**DOI:** 10.3390/ijerph20032627

**Published:** 2023-02-01

**Authors:** Carla Semir-González, Rodrigo Ferrer-Urbina, Carolina Suazo-Navarro, Catalina Flores-Denegri, Darinka Bolados, Joaquín Rosales, Geraldy Sepúlveda-Páez

**Affiliations:** Escuela de Psicología y Filosofía, Universidad de Tarapacá, Arica 1020000, Chile

**Keywords:** chronic non-communicable diseases, obesity, university students, healthy behaviors, eating habits, physical exercise

## Abstract

Chronic non-communicable diseases (NCDs) are a public health problem that affect the quality of life and well-being of people, especially the youth, who have been identified as a high-risk population. Physical inactivity is a key risk factor for NCDs, and an unhealthy diet is a significant driver of NCDs. On the other hand, physical exercise and healthy habits are effective methods of prevention. Although there are scales that measure different behaviors related to NCDs, most of them have been developed in another language (e.g., English) or only focus on one aspect of NCDs. The present study aimed to develop a scale to assess healthy behaviors (i.e., healthy eating and physical exercise) in Spanish-speaking university students, using an instrumental design, with a sample of 369 Chilean university students between 18 and 25 years of age. The results presented show evidence of validity through an exploratory structural equation model (ESEM), reliability estimation through McDonald’s omega and Cronbach’s alpha, evidence of invariance by sex, and evidence of validity in relation to other variables with an SEM model. It is concluded that the Healthy Behavior Scale, consisting of nine items to measure healthy eating and physical exercise, is a brief instrument with evidence of reliability and validity (CFI = 0.998; TLI = 0.995; and RMSEA = 0.063) for application in a Spanish-speaking university population, offering potential applications in research instruments, screening studies, and the development of new studies for other contexts.

## 1. Introduction

According to the World Health Organization (WHO), chronic non-communicable diseases (NCDs) are responsible for 74% of deaths worldwide, mainly through the development of heart disease, cancer, respiratory diseases, and diabetes [1]. NCDs are associated with mental health problems, such as depression and anxiety, which affect people’s quality of life and well-being, and, in addition, impact the global economic problem of healthcare [2]. In the case of Chile, NCDs represent the leading cause of death [3], of which cardiovascular diseases (CVDs), cancer, digestive diseases, chronic renal disease, and diabetes are predominant [4].

Although studies on healthy behaviors in higher education are scarce, university students have been identified as a population at high risk for NCDs [5,6,7], since university life brings with it behavioral changes associated with the transition from adolescence to young adulthood, including changes in eating habits and physical exercise [5,7,8] such as prolonged fasting and a preference for high-fat foods [6,7].

On the one hand, there are various factors associated with NCDs. The most studied risk factors are associated with unhealthy eating habits, [4,9] tobacco and alcohol consumption [4], and physical inactivity [9], all of which act, in part, because they are strongly associated with overweight and obesity [4,10,11,12]. On the other hand, some factors substantially reduce the risk of NCDs, such as healthy eating and moderate physical exercise [11].

Amongst the adult population, healthy eating refers to the habitual behavior of balanced consumption of foods rich in nutrients such as complex carbohydrates, fruits and vegetables, and unsaturated fats, determined by the individual characteristics of individuals in both their physical and psychosocial composition [1]. There is plenty of evidence on the health and quality of life benefits of a healthy diet and the risks of an unhealthy diet [2,7,8,13], which is strongly associated with overweight and obesity, the latter being responsible for reducing life expectancy, leading to the development of NCDs such as diabetes, cardiovascular diseases, and cancer [4,6,10,14].

Physical exercise refers to the body’s movement that leads to energy consumption, such as daily commuting and recreational or sporting activities. Recommended physical exercise considers regular activity subject to the particular needs of individuals and determined by their life cycle and biopsychosocial characteristics [1]. In addition, physical exercise brings multiple health benefits to individuals, such as reduced inflammation and improved muscle composition and cardiac function, among others, which together lead to disease prevention [15], while its deficit, which is common in modern society [4], is the major risk for the development of NCDs, as it promotes an unhealthy diet and is associated with metabolic risks, such as metabolic syndrome and cardiovascular diseases [5,7,16].

Assessment instruments with evidence of validity and reliability are required to support inferences and interpretations of observed scores and provide the minimum ethical safeguards to support the conclusions derived from the measurement process in order to study eating behaviors and physical exercise for the diagnosis and development of evidence-based interventions [17].

In the literature, there is evidence of different instruments for assessing levels of eating habits and physical exercise [18,19,20,21], and they are mostly developed in WEIRD (i.e., Western, Educated, Industrialized, Rich, and Democratic) countries [22], which may pose a threat to the representativeness of the instruments for non-WEIRD populations. However, in both WEIRD and non-WEIRD populations, no instruments have been found with evidence of validity for the university population, nor with the main focus of the present study [23,24], that allow simultaneous and brief assessment of eating behaviors and physical exercise.

Therefore, the present study aims to develop a brief self-report measure of healthy behaviors in Spanish-speaking university students with validity evidence.

## 2. Materials and Methods

### 2.1. Design and Participants

This study had an instrumental cross-sectional design [25]. The sample was non-probabilistic by availability [26]. The minimum sample size (n > 350) was established from a Monte Carlo simulation of 2 latent variables and 16 items to achieve a power estimation of 0.99, using PWRSEM [27].

The sample was composed of 369 university students from the city of Arica and other regions of the northern regions, where 64.5% (n = 238) identified themselves as female and 35.55% (n = 131) as male, with a mean age of 23.9 (SD = 4.04). Additionally, 51.2% of the students reported being overweight or obese.

### 2.2. Instruments

Healthy Behavior Scale: the ad hoc development to assess was as follows: (1) healthy eating, defined as the tendency to consume fruits and vegetables in a balanced way, foods low in sugar, avoidance of dietary excesses, and the maintenance of a balanced diet; (2) physical exercise, defined as planned, structured, and repetitive physical activity to improve or maintain physical fitness. The items were four-point Likert behavioral statements (from 1 = “Strongly disagree” to 4 = “Strongly agree”).

Initially, a total of forty items were drafted (twenty for healthy eating and twenty for physical exercise), which were evaluated by three expert judges (one judge experienced in psychometrics and two professional judges from the health area) in terms of grammatical adequacy and content. The judges individually evaluated each of the items in a range of 1, 0, and −1. From this review, 25 items were retained whose means were higher than 0, with which an online pilot application was carried out (n = 128). Subsequently, the scale was iteratively debugged, eliminating items with values lower than 0.30 in the rectified homogeneity index, trying to reduce the measurement error until obtaining at least acceptable levels of reliability (ω > 0.80 or α > 0.70) [28]. Finally, the consequent scale was composed of 15 items applied for this study. The psychometric evidence of the final scale is reported in the Section 3.

Demographic information and self-perception of weight were obtained as follows: self-developed questions focused on general demographic data such as sex, age, and city of residence. Additionally, considering that self-perception of being overweight has been found to have a significant relationship with Body Mass Index (BMI) [29], a question was included to assess a respondent’s perception of their weight, as follows: “Do you currently consider yourself (underweight, at your ideal weight, overweight, obese, morbidly obese)?”.

### 2.3. Procedure

Data collection was carried out between June and August 2019. Authorization was initially requested from the Heads of the Health Sciences and Social Sciences Departments of the Universidad de Tarapacá, who authorized voluntary and anonymous applications in class schedules that they established. Subsequently, group applications were made in pencil and paper format in the authorized spaces and times. During the application, each respondent was provided with an informed consent form, where the objectives of the research, rights, the voluntary nature of the application, and the guarantees of anonymity and confidentiality were established. Anonymity was safeguarded by excluding any personal identifiers in the questionnaires and separating the informed consent from the instrument beforehand. The application lasted between 7 and 14 min.

The ethical aspects of this study were evaluated and approved by the Psychology Undergraduate Program Committee of the Universidad de Tarapacá.

### 2.4. Statistical Analysis

To establish evidence of validity based on internal structure, an exploratory structural equation model (ESEM) was performed with OBLIMIN rotation [30] and a robust weighted least squares estimation method (WLSMV), which is robust with non-normal discrete variables [31]. The analysis was performed based on the matrix of polychoric correlations [32]. The overall model fit was evaluated considering the cut-off point recommendations proposed by Schreiber [33] (i.e., CFI > 0.95; TLI > 0.95; and RMSEA < 0.06). In order to obtain a brief scale, it was iteratively refined based on the following criteria: conservation of items with strong factor loadings (λ > 0.5); elimination of redundant items; and elimination of items with strong cross-loadings (>0.3) [34]. Reliability was estimated for each dimension using Cronbach’s alpha and McDonald’s hierarchical omega coefficients in their non-ordinal versions. To evaluate the scale’s stability between people of different sexes, a multigroup ESEM was performed using metric and scalar invariance tests. The drops in CFI below 0.05 and increases in RMSEA below 0.010 were considered invariance criteria [35]. Finally, evidence of validity based on the relationship with other variables was established from a structural equation model using the WLSMV estimation method and the polychoric correlation matrix of the relationship between the scale dimensions and the demographic variable of self-reporting on body weight status. The dichotomization of the overweight, obese, and morbidly obese categories, concerning the others, was performed to create the overweight variable. All analyses were performed using the statistical programs jamovi (9.0) and Mplus (8.2). All null hypothesis statistical testing (NHST) was performed at a critical value below or equal to 0.05.

## 3. Results

Table 1 presents the adjustment of the ESEM measurement models, including both the original version (15 items) and the refined version (9 items).

According to the adjustment criteria recommended in the specialized literature (CFI > 0.95; TLI > 0.95; and RMSEA < 0.06) [33], the original model (M1) is a sufficient explanation for the observed co-variation matrix. However, some items presented weak factor loadings and cross-loadings, so we proceeded to refine the instrument considering the following criteria: (1) selection of strong factor loadings (λ > 0.5); (2) elimination of redundant items; and (3) elimination of items whose cross-factorial loadings were strong (>0.3).

The comparative and absolute indexes (i.e., CFI, TLI, and RMSEA) indicate that the debugged ESEM model (M2) is an adequate population representation of the observed relationships.

Factor loadings, covariances between dimensions, and reliability values for each dimension are presented in Table 2.

The factor loadings show adequate representations (λ ≥ 0.50) of their respective variables, with low levels of cross-factor loadings (λ ≤ 0.3). The structural relationships between healthy eating and physical exercise are moderate (r < 0.3) [36]. The reliability estimates are evidenced to be adequate or, at least, sufficient (α > 0.70) [26], depending on whether McDonald’s omega or Cronbach’s alpha estimation is used.

Table 3 shows the results of the invariance tests in the M2 model (nine-item version) according to the sex of the participants. The metric (Δ CFI = −0.015; Δ RMSEA = 0.006) and scalar (Δ CFI = −0.006; Δ RMSEA = 0.005) models, when compared with the configural model, do not evidence practical changes of adjustment in the differentials (RMSEA and CFI), which indicates that the equivalence between factor loadings and factor intercepts have the same significance between the groups. Therefore, it can be assumed that the measurement indicators behave equivalently between groups.

Table 4 presents the relationships between the latent dimensions of the developed ad hoc scale (i.e., healthy eating and physical exercise) and the dichotomized variables from the self-reported body weight status (ideal weight and overweight).

The proposed model presents an adequate fit, except for the RMSEA, which was slightly off, according to the recommendations proposed by Schreiber [33] (CFI = 0.998; TLI = 0.995; and RMSEA = 0.063).

According to the observed relationships, both healthy eating and physical exercise were directly related to ideal weight (γ = 0.188, *p* = 0.018 and γ = 0.226, *p* = 0.004, respectively) and inversely related to overweight (γ = −0.154, *p* = 0.050 and γ = −0.232, *p* = 0.003, respectively). It is important to note that in the case of healthy eating and its relationship with being overweight, only a marginally significant covariation is observed.

## 4. Discussion

This study aimed to develop a scale to assess healthy behaviors (i.e., healthy eating and physical exercise) in Spanish-speaking university students.

The results show that the scale for healthy behavior presents evidence of validity and reliability following the criteria recommended in the literature [33]. The adjustment of the final ESEM model, the magnitude of the factor loadings, and the absence of cross-loadings support the proposed model (with a scale composed of two dimensions and nine items). Additionally, the results obtained in the multi-group ESEM show that interpretation of the scores can be made indistinctly in men and women [35].

As for the evidence of validity based on the relationship with other variables, it is observed that both healthy eating and physical exercise are associated with self-perception of weight, supporting the interpretation of the construct. The nature of these relationships is consistent with the findings of other studies that point to healthy eating and physical exercise as protective factors for overweight [37]. For instance, Lanuza et al. [7] indicate that healthy eating habits (e.g., eating breakfast and consuming homemade food) are related to a better quality of life in the young Chilean population, where 48.8% reported having healthy habits, as well as the same percentage obtained in this study who reported having an ideal weight, which was related to healthy habits (e.g., eating habits and physical exercise). Therefore, this suggests that one in two students has healthy protective habits, which in turn indicates that half of the young population has a risk of being overweight or obese. Furthermore, the findings are inversely consistent with the results of Morales et al. [38] in 2017, who showed that low levels of physical activity are a risk factor for NCDs (e.g., obesity and metabolic syndrome). However, only 36.5% of the young people in that study had this habit, which is lower than those in this study and consistent with the observations of Flores et al. [6] regarding increased risk habits (e.g., unhealthy eating). In the same direction, Durán et al. [39], indicated that the effect of poor diet and poor physical condition were related to overweight and obesity in Chilean university students.

Finally, it is necessary to point out that this study presents some limitations: (1) although non-probabilistic sampling has been indicated as useful for the development of measurement instruments [26], the non-probabilistic nature of the sample may imply that the results are not generalizable, so future research must extend the evidence to new contexts and maximize the generalization potential of this scale. (2) On the other hand, in this study, we analyzed the relationships with self-reported measures of weight perception, and as we have already pointed out, the scale has two dimensions, so it would be necessary to contrast it with other measurement instruments frequently used in the literature (e.g., HEPASEQ) [20] or with the incorporation of other measures such as weight, height, or body mass index reported by each participant, as well as other health indicators such as chronic diseases and comorbidities, in order to extend the psychometric evidence. (3) Lastly, the present study did not include information related to the lifestyle or health condition of the participants, which is information that would be relevant in future studies.

Despite these limitations, we consider that the present study provides a brief assessment instrument of easy application, which could be included in different research developments that seek to know and understand the eating behaviors and physical exercise of people in order to improve living conditions from a nutritional perspective. In addition, having the instrument available for the Spanish-speaking population allows for a much more accurate understanding of the specific behaviors and beliefs that young people have regarding eating behaviors and physical exercise.

## 5. Conclusions

The contributions of this study to research lie in the ability of the scale to measure two healthy behaviors (i.e., healthy eating and physical exercise) with a brief structure and fast application.

This scale can be incorporated into the bases of other instruments, screening studies, and the evaluation of the impact of health promotion activities or other interventions in order to contribute to generating evidence-based actions, promoting greater efficiency of resources, and, above all, the development of effective actions that translate into a better quality of life and life expectancy of students, who at older ages could manifest NCDs.

## Figures and Tables

**Table 1 ijerph-20-02627-t001:** Fit indexes for ESEM, single-group.

Models	Par	χ^2^	df	RMSEA	90% CI	CFI	TLI	SRMR
M1	74	182.788	105	0.062	[0.050, 0.073]	0.991	0.987	0.034
M2	44	46.739	4	0.063	[0.040, 0.086]	0.998	0.995	0.018

*Note*: M1 = ESEM 15 items; M2 = ESEM 9 items; χ^2^ = chi-square; df = degree of freedom; RMSEA = root-mean-square error of approximation; CI = 90% confidence interval; CFI = comparative fit index; TLI = Tucker–Lewis index; SRMR = standardized root-mean-square residual.

**Table 2 ijerph-20-02627-t002:** Standardized factor loadings, factorial covariates, and reliability coefficients for each dimension.

Healthy Behavior Scale	HE	PE
*Healthy eating*		
Evito comer en exceso (I avoid overeating)	**0.747 ****	−0.083
Mantengo una dieta balanceada (I maintain a balanced diet)	**0.757 ****	0.086
Suelo consumir alimentos bajos en azúcar (I tend to eat foods low in sugar)	**0.690 ****	−0.034
Procuro comer frutas y verduras a diario (I try to eat fruits and vegetables daily)	**0.679 ****	0.003
Evito consumir bebidas azucaradas (I avoid sugary drinks)	**0.739 ****	0.015
*Physical exercise*		
Realizo ejercicio constantemente (I exercise constantly)	0.005	**0.954 ****
Me ejercito todas las semanas (I exercise every week)	0.000	**0.934 ****
Procuro mantener rutinas de ejercicio (I try to maintain exercise routines)	0.145	**0.819 ****
Hago actividad física dos o más veces por semana (I exercise two or more times a week)	−0.070	**0.992 ****
Factorial covariations	**0.417 ****	
Alfa (α)	0.806	0.940
Omega (ω)	0.0808	0.940

*Note*: factor loadings >0.4 are in bold; ** *p* < 0.01; HE = healthy eating; PE = physical exercise.

**Table 3 ijerph-20-02627-t003:** Multiple-group CFA.

	Par	χ^2^	df	RMSEA	90% CI	CFI	TLI	SRMR	Δ CFI	Δ RMSEA
Configural invariance	70	70.885	38	0.069	0.043–0.094	0.983	0.968	0.026	–	–
Metric invariance	56	89.345	52	0.063	0.040–0.085	0.981	0.973	0.046	−0.015	0.006
Scalar invariance	49	107.902	59	0.068	0.047–0.088	0.975	0.969	0.057	−0.006	−0.005

*Note*: χ^2^ = chi-square; df = degree of freedom; RMSEA = root-mean-square error of approximation; CI = 90% confidence interval; CFI = comparative fit index; TLI = Tucker–Lewis index; SRMR = standardized root-mean-square residual; CMs = comparisons between models; Δ χ^2^ = change in chi-square; Δ df = change in degree of freedom; Δ CFI = change in comparative adjustment index; Δ RMSEA = change in the error of the mean square of the approximation root.

**Table 4 ijerph-20-02627-t004:** Association between the latent variables of the healthy behavior scales and the self-reported variable on body weight status.

Dimensions	Ideal Weight	Overweight
Healthy eating	0.188 *	−0.154
Physical exercise	0.226 **	−0.232 **

*Note*: ** *p* < 0.01; * *p* < 0.05.

## Data Availability

The data presented in this study are available in this article (and Appendix A).

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
