# Peer review of "Development of a Scale to Measure Healthy Behaviors in Spanish-Speaking University Students"

_ijerph, 2023, doi:10.3390/ijerph20032627_

Round 1

Reviewer 1 Report

Dear authors:

This article presents a very interesting and relevant research question on developing a scale to measure healthy behaviours in university students.

However, some issues need to be clarified in order to better understand and apply the results found.

The article presented meets all the formal requirements and is of sufficient scientific quality to be accepted for publication.

The article has all the components expected of a scientific publication. The sections are clearly defined and developed.

The objectives and questions posed in the article are clearly defined and have been resolved one by one throughout the paper. In order to achieve these objectives, an appropriate and up-to-date methodology has been used throughout, which is explained clearly and in detail in the text. In this way, it has been possible to draw conclusions from the results, which are relevant to the area of work in which this research is framed. The starting theory is connected with the data obtained in the study. All the data necessary to justify the conclusions reached are presented. The results obtained are adequately presented, understandable, useful and meaningful for future research in the field. The discussion is well developed and the novelty of the results is clear. The conclusions are clearly and concisely written and are coherent with the objectives set and the results obtained.

The article demonstrates an adequate knowledge of the relevant literature in the field and cites an appropriate range of bibliographic sources. he references used to support the research objectives are supported by the latest research.

It is an up-to-date article in terms of the knowledge available in the field of study so far, reflected in a very up-to-date bibliography, has a high scientific quality, is very well elaborated and very well structured in the report.

The methodology is adequate.

 In conclusion, the article is well written, easy to understand by the readers and the results are encouraging for further application in the field of the use of Social Networking and its relationship with orthorexia nervosa. Therefore, it is reported that this article meets the necessary requirements for publication in the journal.

Major considerations to be rectified:

1.- How did you carry out the calculation of the study sample? You could include a text on how it was done.

2.- You mention that 51.2% of respondents reported being overweight or obese. Is this data self-reported by the subjects? I suggest that, if so, you look for literature demonstrating the strength of this self-reported data, and based on what criteria they are classified in this variable.

3.- The informed consent that was given to participants is requested.

4.- The project number and Ethics Committee approval document is requested.

5.- It is advised that Table 2 be translated into English.

6.- It is recommended that a section on the strengths and weaknesses of the study be included in the study.

Kind regards

Author Response

Reviewer 1: 
Main considerations to be rectified:
 1.- How did you perform the calculation of the study sample? You could include a text on how it was done.
 Answer: We appreciate the observation and have incorporated this information in the following section: 2.1. Design and Participants: "Instrumental cross-sectional design [25]. The sample was non-probabilistic by availability [26]. The minimum sample size (n>350) was stablish from a montecarlo simulation of 2 latent variables and 16 items to achieve a power estimation of .99, using PWRSEM [27]".
2.- You mention that 51.2% of the respondents reported being overweight or obese. Are these data self-reported by the subjects? I suggest that, if so, you look for literature that demonstrates the robustness of these self-reported data and according to what criteria they are classified in this variable.
Response:We appreciate the observation we have incorporated the information: 
Demographic information and self-perception of weight: self-developed questions focused on general demographic data such as sex, age, and city of residence. Additionally, as external criteria, a question was included to assess the perception of their weight: "do you currently consider yourself (underweight, at your ideal weight, overweight, obese, morbidly obese)?". Considering the evidence that self-perception of being overweight has a significant relationship with Body Mass Index (BMI) [29].
 3.- We request the informed consent that was given to the participants.
Response: We appreciate the observation, we have incorporated this information as supplementary material. 
 4.- The project number and Ethics Committee approval document are requested.
Response: We appreciate the comment, we have incorporated this information.
 5.- It is recommended to translate Table 2 into English.
Response: We appreciate your comments and have corrected table 2.

6.-It is recommended to include in the study a section on the strengths and weaknesses of the study.
We appreciate your comments and have incorporated more information in this section.

Reviewer 2 Report

The study aimed to develop a scale to assess health behavior in university students. However, the study has methodological limitations:

Major Comments:

Authors described health habits and health problems in the introduction, but the health profile of the sample, such as the occurrence of comorbidities, was not evaluated.

Why were data such as weight and height not collected to calculate the BMI? I understand that this is an analysis of self-perception of weight, however the fact that data for measurement was not collected represents a limitation.

The questionnaire should include questions about the frequency of fruit and vegetable consumption to validate information given to participants.

In my opinion, the lack of sample data makes it difficult to correctly analyze the data. If more data are included and the sample is better described, this study may be published in the future.

Minor comments:

In line 74-75, there are only references without a text explain about the subject. In the references there are some web links that not open.

In line 81, male sample is n=131 and not 238.

In line 195 the sentence is out of context.

The manuscript contains some formations errors throughout the text and should be carefully proofread.

Author Response

The authors described health habits and health problems in the introduction, but the health profile of the sample, such as the occurrence of comorbidities, was not assessed.
Response: we appreciate the comment we have incorporated the above in the limitations section. 
Why were data such as weight and height not collected to calculate BMI? I understand that this is a self-perception analysis of weight, however, the fact that data were not collected for measurement represents a limitation.
Response: We appreciate the observation and agree with it, which is why we have modified the discussion section to note it as a limitation. 
On the other hand, in this study we analyzed the relationship with self-reported measures of weight perception, and as we have already pointed out, the scale has two dimensions, so it would be necessary to contrast it with other measurement instruments frequently used in the literature (e.g. HEPASEQ) [20] or with the incorporation of other measures such as weight, height or body mass index reported by each participant, as well as other health indicators such as chronic diseases and comorbidities, in order to extend the psychometric evidence.
The questionnaire should include questions on frequency of fruit and vegetable consumption to validate the information provided to participants.
Response: 
In my opinion, the lack of sample data makes it difficult to analyze the data correctly. If more data are included and the sample is better described, this study could be published in the future.
Response: 
Minor comments:
In line 74-75, there are only references without a text explaining about the topic. In the references there are some web links that do not open.
Response: Inconsistent references were removed. All references were reviewed, updated and the links were corrected.
On line 81, the male sample is n=131 and not 238.
Response: We appreciate the comment, we have corrected this information.
On line 195 the sentence is out of context.
Response: 
The manuscript contains some formation errors throughout the text and should be carefully revised.

Reviewer 3 Report

Accept after incorporating my suggestions.

Author Response

Reviewer 3: Formatting errors were reviewed and corrected. However, some errors were editorial errors.
 The manuscript entitled: Development of a scale to measure healthy behaviors in Spanish-speaking college students. The article needs improvement: I have attached several considerations that will be beneficial to improve the manuscript. 
Line 3: Indicate which scientific method was used. Development of a scale to measure healthy behaviors in Spanish-speaking college students: ........? 
Response
Line 12: Physical inactivity is a key risk factor for NCDs, and an unhealthy diet is a major contributor to NCDs. Also note two main variables in the research presented. 
Response: The variables discussed in the article are incorporated into the abstract.
Lines 17-18: Define abbreviations (ESEM, SEM). Line 19: The authors did not mention the physical activity variable. Why? 
Response: we appreciate the comment and have made the requested changes. 
Line 46-59: How physical inactivity, and an unhealthy diet, cause NCDs should be explored. How does the youth's lifestyle affect NCDs? How do physical inactivity and an unhealthy diet affect public health? In that context, discuss metabolic risk factors. 
Response: More information on the risks of developing NCDs is incorporated for both physical inactivity and an impoverished diet.
Line 79: reference 25 should be in parentheses. 
Response: The formatting is corrected (this formatting error, I believe, is editorial).
Line 119: The ethics approval number and date need to be included. 
Response: We appreciate this observation and have included this information. 
Line 120: Where in the statistical analysis is the p-value indicated? 
Response: We appreciate the comment we have incorporated this precession in the statistical analysis section. 
Line 165: Table 2. Define the abbreviations AS and EF. 
Response: We appreciate your comments and have corrected table 2.
Line 196: Discussion - In this section, the author should discuss the results and how they can be interpreted from the perspective of previous studies. The results and implications should be discussed in the broadest context, not the briefest. Future lines of research can also be highlighted. 
Response: we appreciate the comment we have made the changes to the paragraph.
Line 210: Discuss in more depth the possible limitations and strengths of the study presented. 
Response: We appreciate the comment and have incorporated more information. 
Line 231: The ethics approval number and date should be included.
Response: We appreciate the comment, we have incorporated this information. 

Round 2

Reviewer 1 Report

Dear authors:

Thank you very much for your kind comments and responses.

I consider the manuscript to be correct for publication.

Best regards

Author Response

We appreciate the opportunity given to our study titled: “Development of a scale to measure healthy behaviors in Spanish-speaking university students”. The revisions and comments provided by the editor and reviewers are undoubtedly a great contribution that allowed us to make substantial improvements in the study.

          In the following, we present the responses to the reviewers' comments, trying to address each of the points identified, explaining how we have incorporated their comments into the revised manuscript and, in the few cases where we have not been able to do so, we have tried to explain them in detail.

Comments and Suggestions for Authors

I appreciate that the authors have made improvements to the article. However, information about the participants lifestyle and health conditions should be included in future studies.

Response: We appreciate your suggestion. We have incorporated, in the limitations of our study, the need to incorporate the lifestyle and health condition of the participants in future studies.

In addition, in the discussion section, the results need to be better discussed, relating them to data from previously published studies.

Response: Grateful for your observation, we have strengthened the discussion by comparing results from previous studies.

I suggest that you include the complete questionnaire in the supplementary material.

Response: We appreciate the suggestion, we have included the complete form.

Most grateful

Sincerely.

 Geraldy Sepúlveda, Carla Semir & Rodrigo Ferrer.

Reviewer 2 Report

I appreciate that the authors have made improvements to the article. However, information about the participants lifestyle and health conditions should be included in future studies.

In addition, in the discussion section, the results need to be better discussed, relating them to data from previously published studies.

I suggest that you include the complete questionnaire in the supplementary material.

Author Response

(The authors gave the same response as above.)
